# Minor introns are embedded molecular switches regulated by highly unstable U6atac snRNA

Ihab Younis[1], Kimberly Dittmar[1], Wei Wang[2], Shawn W Foley[1], Michael G Berg[1], Karen Y Hu[1], Zhi Wei[2], Lili Wan[1], Gideon Dreyfuss[1]*

[1]Department of Biochemistry and Biophysics, Howard Hughes Medical Institute, University of Pennsylvania School of Medicine, Philadelphia, United States; [2]Department of Computer Science, New Jersey Institute of Technology, Newark, United States

**Abstract** Eukaryotes have two types of spliceosomes, comprised of either major (U1, U2, U4, U5, U6) or minor (U11, U12, U4atac, U6atac; <1%) snRNPs. The high conservation of minor introns, typically one amidst many major introns in several hundred genes, despite their poor splicing, has been a long-standing enigma. Here, we discovered that the low abundance minor spliceosome's catalytic snRNP, U6atac, is strikingly unstable (t½<2 hr). We show that U6atac level depends on both RNA polymerases II and III and can be rapidly increased by cell stress-activated kinase p38MAPK, which stabilizes it, enhancing mRNA expression of hundreds of minor intron-containing genes that are otherwise suppressed by limiting U6atac. Furthermore, p38MAPK-dependent U6atac modulation can control minor intron-containing tumor suppressor PTEN expression and cytokine production. We propose that minor introns are embedded molecular switches regulated by U6atac abundance, providing a novel post-transcriptional gene expression mechanism and a rationale for the minor spliceosome's evolutionary conservation.

*For correspondence: gdreyfuss@hhmi.upenn.edu

Competing interests: The authors declare that no competing interests exist.

## Introduction

Small nuclear RNAs (snRNAs) are a class of non-coding RNAs (ncRNAs) that function in pre-mRNA splicing (*Wahl et al., 2009*) and in mRNA length regulation (telescripting) (*Kaida et al., 2010*; *Berg et al., 2012*). The vast majority of introns (>200,000) in complex eukaryotes are spliced by the major spliceosome (*Wahl et al., 2009*), consisting of U1, U2, U4, U5 and U6 snRNPs. However, ~700 minor introns, also known as U12 introns, are spliced by the much less abundant (<1%) minor spliceosome, consisting of U11, U12, U4atac, U5 and U6atac snRNPs (*Hall and Padgett, 1996*; *Tarn and Steitz, 1996*; *Patel and Steitz, 2003*). Despite their low occurrence, minor introns, typically one amidst many major introns, are found in genes essential for diverse cellular processes (*Burge et al., 1998*; *Sheth et al., 2006*). Their splice site sequences, position and the genes in which they reside are all highly conserved (*Basu et al., 2008*; *Turunen et al., 2012*). However, previous data suggested that minor introns are generally poorly spliced, and therefore the purpose of the minor splicing pathway has been a long-standing mystery, as it has been difficult to rationalize their remarkable conservation and function in splicing alone. The importance of the minor spliceosome has been recently underscored by reports that mutations in U4atac cause microcephalic osteodysplastic primordial dwarfism type I (MOPD I) or Taybi-Linder syndrome (TALS) (*Abdel-Salam et al., 2011*; *Edery et al., 2011*; *He et al., 2011*). We have previously shown that perturbations in the repertoire of both major and minor snRNPs cause widespread transcriptome and splicing defects in spinal muscular atrophy (*Zhang et al., 2008*).

Using a ncRNA microarray that we devised and mRNA sequencing (RNA-Seq), we discovered that U6atac, the minor spliceosome's low abundance catalytic snRNA, also has a very short half-life and can

**eLife digest** The central dogma of biology states that genetic material, DNA, is transcribed into RNA, which is then translated into proteins. However, the genes of many organisms have stretches of non-coding DNA that interrupt the sequences that code for protein. These non-coding sequences, which are called introns, must be removed, and the remaining sequences—which are called exons—must then be joined together to produce a messenger RNA (mRNA) transcript that is ready to be translated into protein.

The process of removing the introns and joining the exons is called splicing, and it is carried out by a molecular machine called the spliceosome. However, in addition to containing typical ('major') introns, several hundred human genes also contain a single 'minor' intron, and a minor spliceosome is needed to remove it. Minor introns occur in many highly conserved genes, but they are often inefficiently spliced. This means that the resulting mRNA transcripts may not be translated into proteins—which is puzzling given that these proteins perform important roles within the cell.

The major and minor spliceosomes are composed of proteins and small non-coding RNA molecules (which, as their name suggests, are never translated in cells). Now Younis et al. shed new light on the minor spliceosome by showing that a small non-coding RNA molecule known as U6atac, which catalyzes the removal of introns by the minor spliceosome, is highly unstable in human cells. This means that U6atac is a limiting factor for the splicing of minor introns—a process that is already limited by the very low abundance of the minor spliceosome under normal conditions. However, Younis et al. found that this bottleneck could be relieved by halting the degradation of U6atac. Experiments showed that U6atac can be stabilized by a key signaling molecule, a protein kinase (called p38MAPK), which is activated in response to stress. The resulting higher levels of U6atac promoted splicing of the introns in its target mRNA transcripts, and also modulated various signaling pathways in the cells.

Together, these results imply that the minor spliceosome is used as a valve that can help cells to adapt to stress and other changes. Moreover, by helping to translate mRNA transcripts that are already present in cells, it enables proteins to be produced rapidly in response to stress, bypassing the need for a fresh round of transcription.

be rate limiting. We found a wide variation of minor intron dependence on U6atac level and that mRNA production from ~200 minor intron-containing genes are strongly suppressed in HeLa cells under normal growth conditions. Importantly, we show that U6atac level can be rapidly up-regulated by activated p38MAPK, which results in enhanced splicing of minor introns and increased expression of these mRNAs. Among these is the tumor suppressor, PTEN, which we show buffers cell response to stress by modulating TNF-α and IL-8 cytokine production. We propose that minor introns are control switches that are regulated by U6atac abundance, providing a novel gene expression regulation mechanism and explaining the retention of the minor splicing pathway during evolution.

## Results

### U6atac snRNA turns over rapidly

A deeper understanding of ncRNAs' diverse and fundamental functions requires quantitative, high-throughput, and systematic profiling in various cell types and physiological conditions. Here, we designed a custom array with probes complementary to 129 human ncRNAs, including snRNAs, rRNAs, snoRNAs, scaRNAs, tRNAs and others (*Supplementary file 1A*), but excluding miRNAs for which commercial microarrays exist (*Davison et al., 2006*). Total RNAs were directly labeled, alleviating the need for amplification and reducing the distortion often caused by this step (*Hiley et al., 2005*). Hybridization conditions were optimized and the reproducibility between biological replicates is shown in *Figure 1—figure supplements 1 and 2*. Our method allows for the rapid measurement of many ncRNAs over a large dynamic range with high sensitivity and specificity, making it a powerful tool for global analysis of ncRNA levels.

The ncRNA microarray was used here to monitor global changes in the levels of ncRNAs after transcription and translation inhibition and cell cycle arrest. Interestingly, Pol II transcription inhibition

by Actinomycin D (ActD), as well as by the structurally and chemically unrelated Pol II inhibitor 5,6-Dichloro-1-β-D-ribofuranosylbenzimidazole (DRB), led to a rapid and strong decrease in the levels of three ncRNAs transcribed by Pol III: U6atac and U6 snRNAs, and vault RNAs (*Figure 1A*). Previous studies have shown co-dependence of Pol III transcription on Pol II, which binds upstream of Pol III at the promoter of these ncRNAs (*Listerman et al., 2007*). The decrease in vault RNAs after ActD treatment has been previously observed in mouse cells (*Kickhoefer et al., 2001*), and U6 was noted to have a shorter half-life compared to other major snRNAs (*Fury and Zieve, 1996*). However, this is the first report of U6atac's fast turnover. We have not found significant sequence similarity among these ncRNAs to explain their fast turnover. Real-time qPCR validated the short half-life of U6 and U6atac, but not U5, for which the microarray did not predict a change (*Figure 1B*). In comparison, HeLa cells arrested in mitosis using nocodazole, which causes transcription by all three types of polymerases to cease, showed a moderate U6atac decrease, whereas translation inhibition with cycloheximide (CHX) caused no significant changes in the levels of the ncRNAs tested (*Figure 1A*). In cells, U6atac mostly

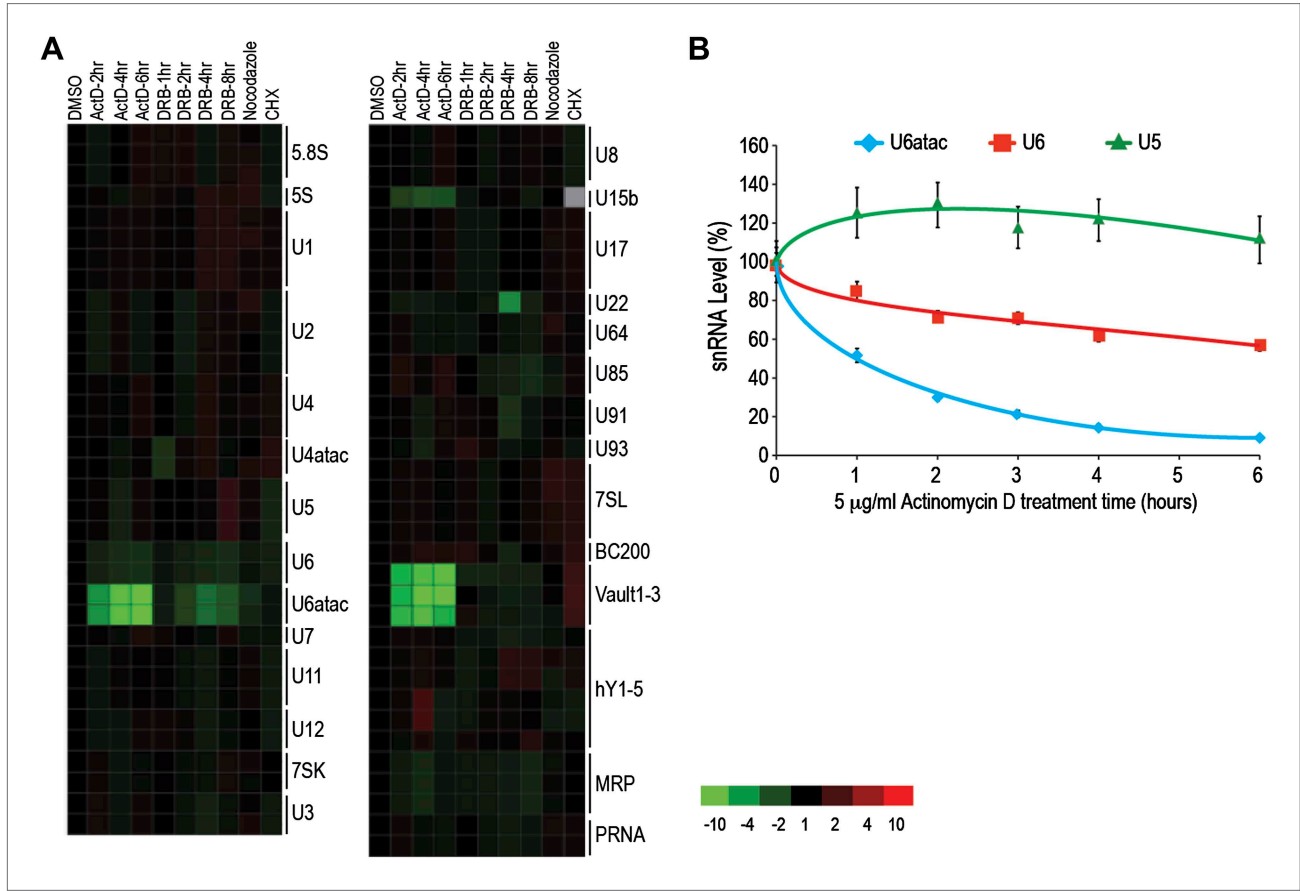

**Figure 1**. U6atac is an extremely unstable ncRNA. (**A**) The heat map summarizes data for HeLa cells treated with inhibitors of several cellular processes, including the transcription inhibitors Actinomycin D (ActD) for 2–6 hr or DRB for 1–8 hr, the cell cycle inhibitor nocodazole for 18 hr, and the translation inhibitor cycloheximide (CHX) for 16 hr. A color scale bar for heat maps (−10 to +10 fold change) is indicated. (**B**) Real time qPCR verification of U5, U6 and U6atac snRNA level after ActD treatment. Absolute snRNA levels were measured in triplicates and normalized to 5S rRNA. Error bars represent standard deviations of three replicates.

The following figure supplements are available for figure 1:

**Figure supplement 1**. Hybridization of total RNA and in vitro transcribed RNA standards.

**Figure supplement 2**. ncRNA microarray is highly reproducible.

**Figure supplement 3**. U6atac, not U4atac, is rate limiting for di-snRNP association.

exists as part of U4atac/U6atac di-snRNP or U4atac/U6atac/U5 tri-snRNP complexes. To test whether U6atac instability is rate limiting for U4atac/U6atac di-snRNP formation, we immunoprecipitated the di-snRNP specific protein p110/SART3 and measured the associated U4atac/U6atac by real-time PCR in control or ActD-treated cells. This showed that ActD decreased U4atac level by ~20% in total RNA, whereas the U4atac/U6atac di-snRNP formation was reduced by >80% (*Figure 1—figure supplement 3*). These results demonstrate the fast turnover and vulnerability of U6atac to transcriptional inhibition, which makes it rate limiting for di-snRNP formation.

## Nascent transcriptome sequencing reveals that U6atac level limits mRNA synthesis from hundreds of minor intron-containing genes

To determine the effects of U6atac inactivation on the transcriptome, we used an antisense morpholino oligonucleotide (AMO) that targets the 5′ end of U6atac snRNA to rapidly inhibit its activity in cells (*Konig et al., 2007*). RNase H protection using an antisense DNA probe complementary to the same sequence confirmed the binding of the AMO to U6atac snRNP and its efficient interference following an 8 hr transfection with 15 nmole of U6atac AMO (*Figure 2A*), which was used for all subsequent

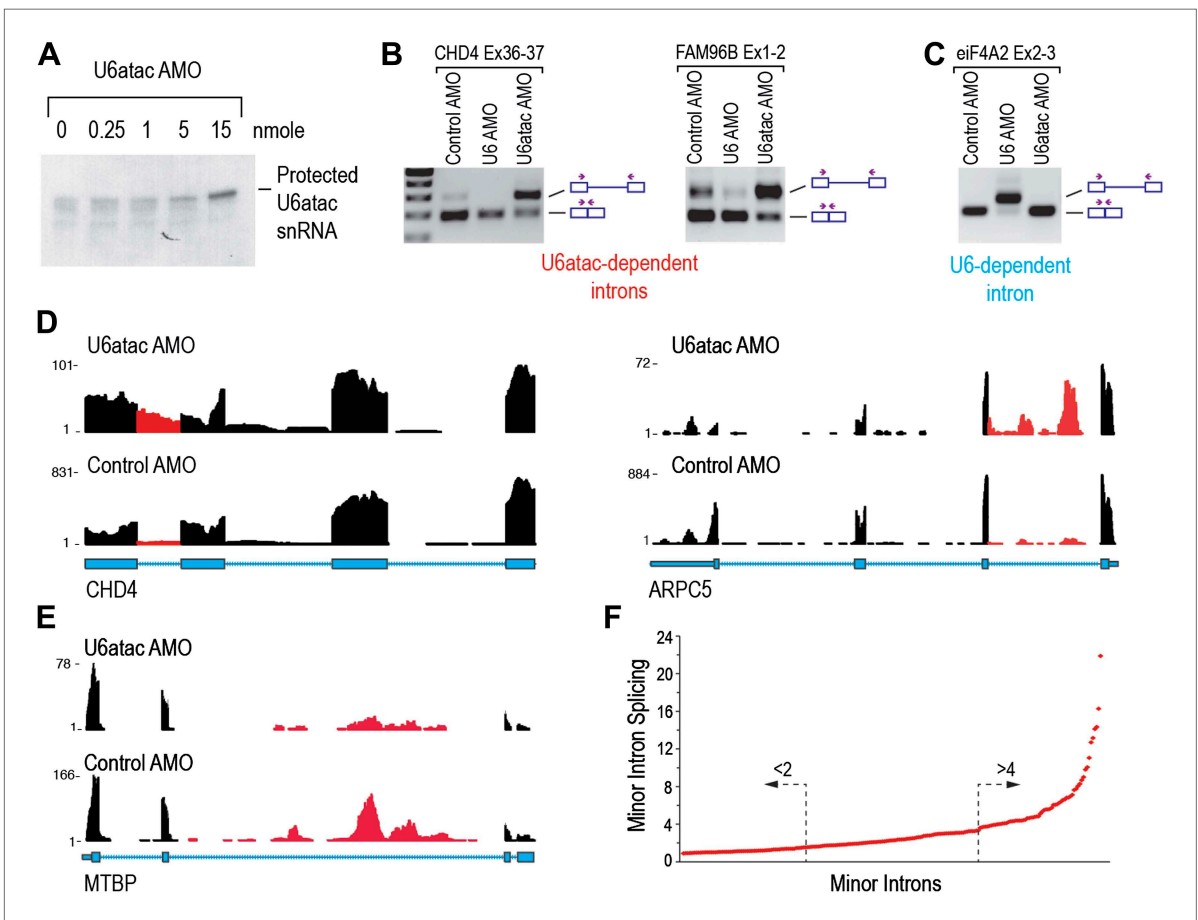

**Figure 2**. U6atac inactivation shows a wide range of minor intron splicing dependence on U6atac level. (**A**) RNase H protection of U6atac snRNA in total cellular RNA after 8 hr treatment with U6atac AMO showing efficient U6atac protection at 15 nmole U6atac AMO. The same RNA sample (15 nmole U6atac AMO) was used for RT-PCR of representative minor (**B**) and major (**C**) introns. The diagrams to the right of the gel indicate spliced or unspliced RNA. Arrows show position of the primers used. (**D**) Representative view on UCSC genome browser for two genes containing highly efficient minor introns (red reads). Numbers on the left represent peak reads count. Gene structures are depicted in blue boxes (exons) and lines (introns). (**E**) Same as in (**D**) for low efficiency minor intron. (**F**) The splicing index is plotted for each expressed minor intron in HeLa cells. The dashed arrows point to minor introns with the lowest (<twofold) and highest (>fourfold) splicing indices.

The following figure supplements are available for figure 2:

**Figure supplement 1**. Functional knockdown of U6atac affects a large number of genes.

experiments. The specificity and efficiency of the AMOs were verified by RT-PCR showing that U6atac AMO, but not control AMO or U6 AMO, resulted in inhibition of minor intron splicing (*Figure 2B*), whereas U6 AMO inhibited splicing of only the major introns (*Figure 2C*).

To identify transcriptome and splicing changes that would be most readily detected in nascent transcripts, we metabolically labeled RNAs in HeLa cells with 4-thiouridine for the last 2 hr of AMO treatment, then selected and sequenced only nascent polyadenylated transcripts produced in the time window during which U6atac was functionally inhibited. More than 120 million RNA-seq reads were obtained for each sample, with ~70% of these mapping to 9799 and 8397 genes in control and U6atac AMO-treated cells, respectively (*Table 1*). Cufflinks (*Trapnell et al., 2012*) was used to assemble transcripts, estimate their abundance, and score differential expression.

Focusing first on predicted minor introns (~700) in the minor intron database (U12DB) (*Alioto, 2007*), we detected 429 that passed the threshold for expressed genes of fragments per kilobase per million mapped reads (FPKM) > 1 in control AMO. Interestingly, 206 (48%) of these were expressed at low levels in control and severely down-regulated (FPKM < 1) in U6atac AMO, suggesting that limiting amount of U6atac suppresses splicing and expression of these genes even under normal growth conditions, and that their unspliced pre-mRNAs are very unstable. On the other hand, unspliced, yet stable pre-mRNAs for the remaining 223 introns were detected. To rank these minor introns according to their relative dependence on U6atac level, we calculated the splicing index for each intron as the ratio of Y/X whereas: X = the reads count in a given intron normalized to the reads count in the surrounding exons in control AMO. Y = the reads count for the same intron normalized to the reads in the surrounding exons in U6atac AMO. This showed that 44 (20%) out of these 223 have little to no intronic reads in control but significant intron accumulation in U6atac AMO (*Figure 2D*; CHD4, ARPC5). For these minor introns, the level of U6atac snRNP is sufficient for splicing catalysis under normal growth conditions (control AMO), but they are extremely sensitive to U6atac decrease. In contrast, 81 (36%) minor introns were poorly spliced, with their unspliced pre-mRNAs clearly detectable in control and showing minimal additional response to U6atac inactivation (*Figure 2E*; MTBP). For these introns, U6atac snRNP is limiting under normal growth conditions and they are predicted to splice more efficiently when U6atac level increases. Taken together, these data show a strict dependence on U6atac abundance for minor intron splicing, which actually varied over a wide range (>20-fold; *Figure 2F*) and thus challenges the accepted view that minor introns typically splice poorly. This wide range could not be explained by analysis of the splice sites (AT-AC vs GT-AG) or their surrounding sequences (data not shown). A list of minor introns with highest and lowest U6atac-dependent splicing indices is shown in *Supplementary file 1B*.

Comparison of mRNA levels (*Table 1* and *Figure 2—figure supplement 1*) showed that U6atac AMO caused >twofold down-regulation in 2088 genes, including all 429 expressed minor intron genes. Only three genes were up-regulated. Splicing analysis revealed 657 alternative splicing changes resulting from U6atac inactivation (*Figure 3A*), including all 223 minor introns that showed various levels of splicing inhibition. Importantly, the major introns in these same transcripts were spliced normally. In a small number of minor intron-containing genes (<20) U6atac inactivation elicited alternative major intron splicing from cryptic 5′ and 3′ splice sites around the minor introns (e.g., C11ORF10 and ZNF207; *Figure 3B*). This resulted in mRNA isoform switching without overall transcript level down-regulation. Interestingly, while these effects are likely due to misplicing as a result of U6atac inhibition, they nevertheless resemble the previously described twintrons that utilize either the minor or major spliceosome by switching between adjacent U11-U12 or U1-U2 splice sites, and play an important role in *Drosophila* development (*Scamborova et al., 2004*), but has not been previously identified in human cells. U12 AMO, tested on select minor introns (*Figure 3*),

**Table 1.** Summary of the RNA-seq statistics

| | Control AMO | U6atac AMO |
|---|---|---|
| Platform | Illumina HiSeq2000 | Illumina HiSeq2000 |
| Read length | 100 base | 100 base |
| Single/paired ends | single end | single end |
| Total number of reads | 120,274,499 | 121,794,728 |
| Mapped reads | 81,198,953 | 85,645,962 |
| Expressed genes (FPKM > 1) | 9799 | 8397 |
| Upregulated genes (> fourfold) | ND | 3 |
| Downregulated genes (< fourfold) | ND | 2088 |
| Expressed U12-dependent genes (FPKM > 1) | 429 | 350 |

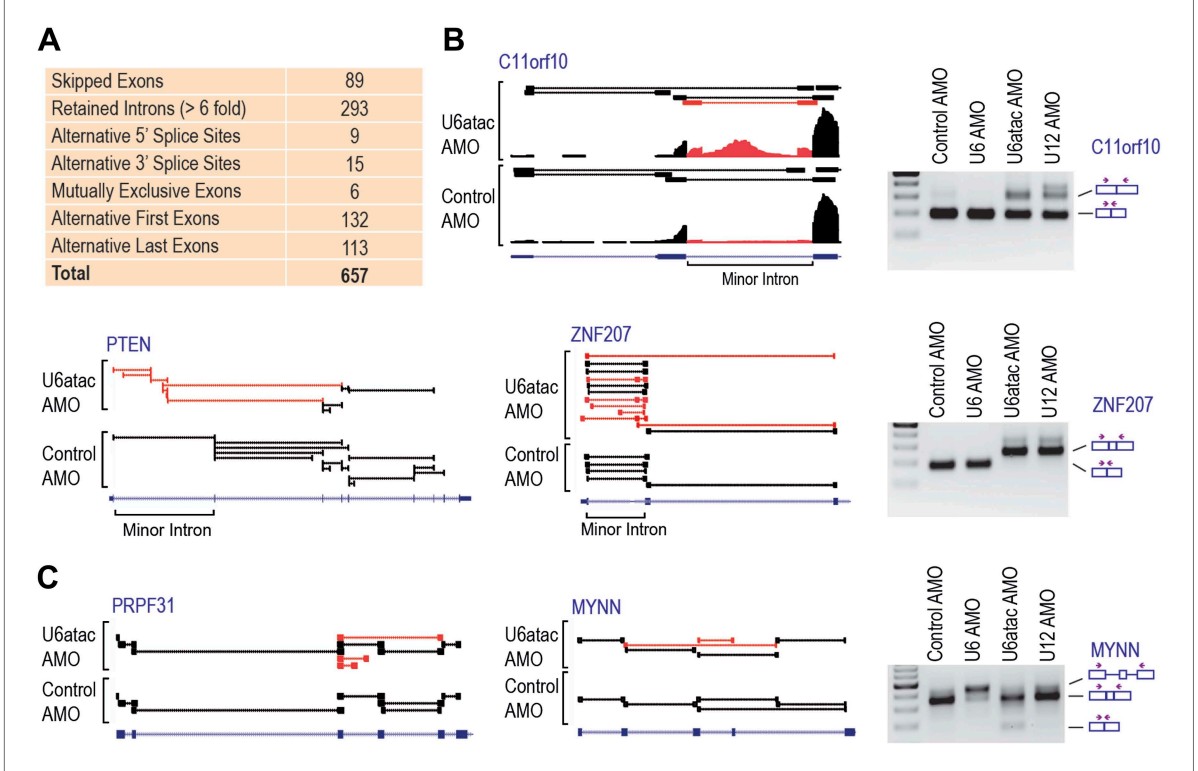

**Figure 3**. Genome-wide analysis of the minor splicing pathway shows widespread transcriptome changes. (**A**) A summary of the splicing effects in the U6atac AMO treated cells identified using MISO. (**B**) Representative minor intron-containing genes showing alternative splicing pattern changes after U6atac knockdown. Exon junction reads are presented on top. Red junction reads are unique to U6atac AMO sample. RT-PCR confirmations for U6atac AMO as well as U6 and U12 knockdown for comparison are also shown. Arrows show position of the primers used, which are located in the exons. (**C**) Representative genes that do not contain a minor intron showing alternative splicing pattern changes after U6atac inactivation.

had the same effects as U6atac AMO, confirming that the U6atac AMO effects we observed result from the minor spliceosome inhibition. Although the effect of U6atac inactivation on expression level and alternative splicing in ~2000 genes that have no minor introns could be secondary, it nevertheless reflects a significant and immediate biological phenomenon. We conclude that U6atac decrease down-regulates or causes splicing change in all minor intron genes and that minor spliceosome inhibition reverberates widely throughout the transcriptome.

## p38MAPK activation increases U6atac level and up-regulates minor intron splicing

As many of the minor intron-containing genes that were significantly affected by U6atac level change have key roles in growth regulation and cell stress signaling (***Supplementary file 1B***), we explored the possibility that U6atac level might change in response to specific cell stress. For this we tested the effect of anisomycin, a potent activator of stress-induced protein kinase, p38MAPK (***Bunyard et al., 2003***; ***Yong et al., 2010***), and observed a ~twofold increase in U6atac level within 4 hours (***Figure 4A***). Because anisomycin activates both p38MAPK and JNK, we pre-treated cells with either SB203580, an inhibitor of p38MAPK activation (***Davies et al., 2000***), or SP600125, an inhibitor of JNK (***Heo et al., 2004***). While SB203580 blocked the anisomycin-mediated U6atac increase, SP600125 had only a small effect (***Figure 4A***), indicating that U6atac increase is predominantly due to p38MAPK activation. To assess whether this increase is transcriptional or post-transcriptional, cells were co-treated with ActD and anisomycin. In contrast to the drastic U6atac decrease after transcription inhibition with ActD (82%), co-treatment with anisomycin resulted in only about half that amount of U6atac decrease (~40%) (***Figure 4B***), suggesting that cell stress-induced U6atac increase is due, at least in part, to U6atac stabilization. Interestingly, the U6atac increase correlated with enhanced minor intron splicing

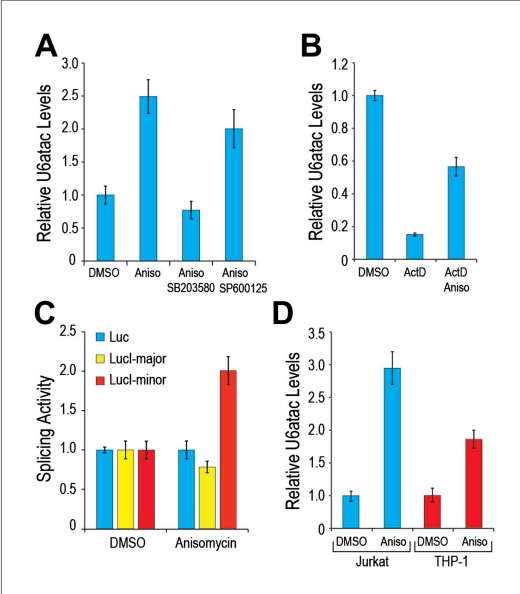

**Figure 4**. p38MAPK cell signaling up-regulates U6atac level and minor intron splicing. (**A**) Real time qPCR of U6atac snRNA after treatment with 1 μg/ml anisomycin (4 hr), a potent p38MAPK activator, in HeLa cells. Cells were treated with 10 μM of p38MAPK inhibitor, SB203580, or JNK inhibitor, SP600125, 30 min prior to anisomycin. (**B**) HeLa cells were treated with 5 μg/ml Actinomycin D (ActD) or ActD plus 1 μg/ml anisomycin (Aniso) or DMSO as control for 4 hr, followed by measurement of U6atac level. (**C**) Luciferase activity of the various splicing reporters after cell treatment with DMSO or anisomycin for 4 hr. (**D**) Real time qPCR of U6atac snRNA after treatment with anisomycin (4 hr) in Jurkat T cells and THP-1 monocytes. Error bars represent the standard deviation of at least three replicates.

and gene expression of a splicing reporter (Lucl-minor) whose expression is strictly dependent on splicing of a minor intron, showing a twofold increase of its expression after anisomycin treatment (**Figure 4C**). In contrast, intronless and major intron-containing reporters (**Younis et al., 2010**) were not affected, indicating that the increased expression is due to enhanced minor intron splicing and not an effect on transcription or translation. A similar U6atac increase was observed in Jurkat T-cells and THP-1 monocytes following anisomycin treatment (**Figure 4D**), indicating that U6atac increase by activated p38MAPK is a general phenomenon and not limited to HeLa cells.

Similarly, p38MAPK activation with anisomycin increased minor intron splicing and mRNA levels of several endogenous genes that contain minor introns (e.g., PTEN and E2F2; **Figure 5A**). Splicing and mRNA levels of eIF3K, whose minor intron splices with high efficiency in control cells, as well as GAPDH and actin, used as controls because they have no minor introns, were unaffected (**Figure 5**). Importantly, anisomycin also enhanced splicing and expression from genes whose levels are suppressed in HeLa under normal growth conditions (**Figure 5B**; e.g., BRMS1L and E2F6). These data are consistent with the idea that limiting U6atac suppresses the expression of several hundred minor intron-containing genes, which can be up-regulated by U6atac increase.

## U6atac level change plays a role in cellular stress response

To determine if the p38MAPK–induced up-regulation of minor intron splicing depends on U6atac increase, we examined, as an example, the splicing of PTEN's minor intron, which increased by up to fivefold after 4 hr of anisomycin treatment (**Figure 6A**). This increase was completely blocked after co-treatment with U6atac AMO, even at a low amount (1 nmole) that only partially inactivates U6atac (**Figure 6A**), confirming that p38MAPK up-regulation of minor intron-containing mRNAs expression is strictly reliant on its ability to increase U6atac level.

We next asked if U6atac level plays a role in specific physiological downstream effects of p38MAPK activation. Whereas p38MAPK activation increases transcription of several key cell physiology modulators, including up-regulation of the cytokines TNF-α and IL-8 (**Figure 6B,C**), which do not contain minor introns (**Chang et al., 2006**), production of these cytokines has been shown to be suppressed by PTEN, a minor intron-containing gene (**Furumoto et al., 2006**; **Furgeson et al., 2010**). We therefore determined if U6atac inactivation affects p38MAPK-induced TNF-α and IL-8. As shown in **Figure 6B,C**, U6atac inactivation increased the levels of p38MAPK-mediated TNF-α and IL-8 production to a much higher level, >twofold compared to control AMO, suggesting that a minor intron-containing gene(s) suppresses production of these cytokines. To test if this is due to loss of PTEN suppression, PTEN was knocked down by siRNA (**Figure 6D**). This resulted in the same effect on p38MAPK-induced TNF-α and IL-8 production as U6atac AMO (**Figure 6**, compare panels 6E and 6F to 6B and 6C), indicating that PTEN expression, which is regulated by U6atac level, plays a role in p38MAPK-induced cytokine production. We note, however, that while PTEN mRNA level was increased by 2–5 fold we have only been able to detect a small (~20%) increase of PTEN protein level by western blot. Nevertheless, previous reports have shown that even very small changes in PTEN level are sufficient to have a strong phenotype (**Alimonti et al., 2010**).

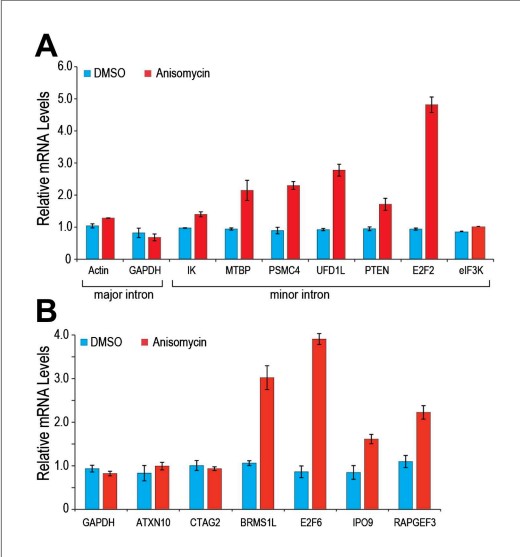

**Figure 5**. p38MAPK activation increases expression of minor intron-containing genes. (**A**) Real time qPCR for a representative set of minor introns with low splicing indices with or without 1 µg/ml anisomycin treatment. All primer sets span the splice junction and thus measure spliced mRNAs. (**B**) Real time qPCR for representative minor intron-containing genes that are normally suppressed in HeLa cells. Error bars represent the standard deviation of three replicates.

Importantly, our findings establish a role for U6atac in regulation of major signaling pathways.

## Discussion

The findings we describe reveal a novel mechanism for gene regulation as well as have important implications for cell signaling, and they provide a new perspective to explain the evolutionary conservation of the minor spliceosome. Previous studies using a small number of model minor introns have led to the perception that minor introns generally splice at a slower rate than major introns and could therefore be rate limiting for the production of mRNAs that contain them (*Patel et al., 2002*; *Singh and Padgett, 2009*). However, there has been no evidence that minor introns could be regulated. Here we discovered that U6atac is highly unstable relative to other snRNAs and its level can be rapidly up- or down-regulated. Furthermore, modulation of U6atac amount produces a corresponding change in the amount of mRNA from many minor intron-containing genes, thus providing a novel post-transcriptional mechanism for regulating minor intron splicing and the expression of the genes that contain them, as depicted in *Figure 7*. Interestingly, in growing HeLa cells ~50% of minor intron-containing genes are suppressed by the limiting U6atac level and produce very low amounts of mRNA. Many others are strongly down-regulated with U6atac decrease because a retained minor intron causes the pre-mRNA to degrade or switch alternative splicing to make a different isoform. However, their splicing and expression can be strongly and rapidly enhanced by an increase in U6atac level, at least in part due to its stabilization by p38MAPK (*Figure 7*). Thus, the apparent sluggishness observed for some minor introns can be explained by the rarity of U6atac, rather than an intrinsic inefficiency of minor introns. Furthermore, the rapid turnover of U6atac after attenuation of either RNA polymerases II or III makes the minor spliceosome a real time sensor of transcriptional activity in cells.

The increased amount of U6atac following p38MAPK activation indicates that U6atac and the many minor intron-containing genes whose expression depends on sufficient U6atac are unexpected downstream targets of p38MAPK. These genes, including PTEN, E2F2, E2F6 and MTBP, play important roles in cell stress physiology such as regulating cytokine production. PTEN, phosphatase and tensin homolog, is a tumor suppressor that plays key roles in cell growth and apoptosis (*Tamguney and Stokoe, 2007*). We show that U6atac and the minor splicing pathway, through regulation of PTEN expression, play an antagonistic role that buffers p38MAPK-induced production of TNF-α and IL-8, two major inflammatory response mediators. TNF-α also functions in a wide range of processes including cell proliferation, differentiation, apoptosis, lipid metabolism and coagulation. IL-8 is a chemo-attractant and a potent angiogenic factor. While activated p38MAPK has been shown to phosphorylate the transcription factor ATF2, which in turn activates PTEN transcription (*Shen et al., 2006*; *Qian et al., 2012*), our data show that PTEN expression can also be regulated post-transcriptionally by the efficiency of splicing of its minor intron. This illustrates a physiological role for minor splicing pathway in regulating cell signaling response.

It has been difficult to rationalize the conservation of minor introns and the minor spliceosome on the basis of splicing alone, as with slight sequence variation this function could have simply been relegated to the major spliceosome. Indeed, despite their early evolutionary origin, minor introns and snRNPs have been lost at multiple points during eukaryotic evolution (*Bartschat and Samuelsson, 2010*), suggesting that, in contrast to the major spliceosome, they are not absolutely essential. We propose a new perspective on the conservation of minor introns and their splicesome, that they function

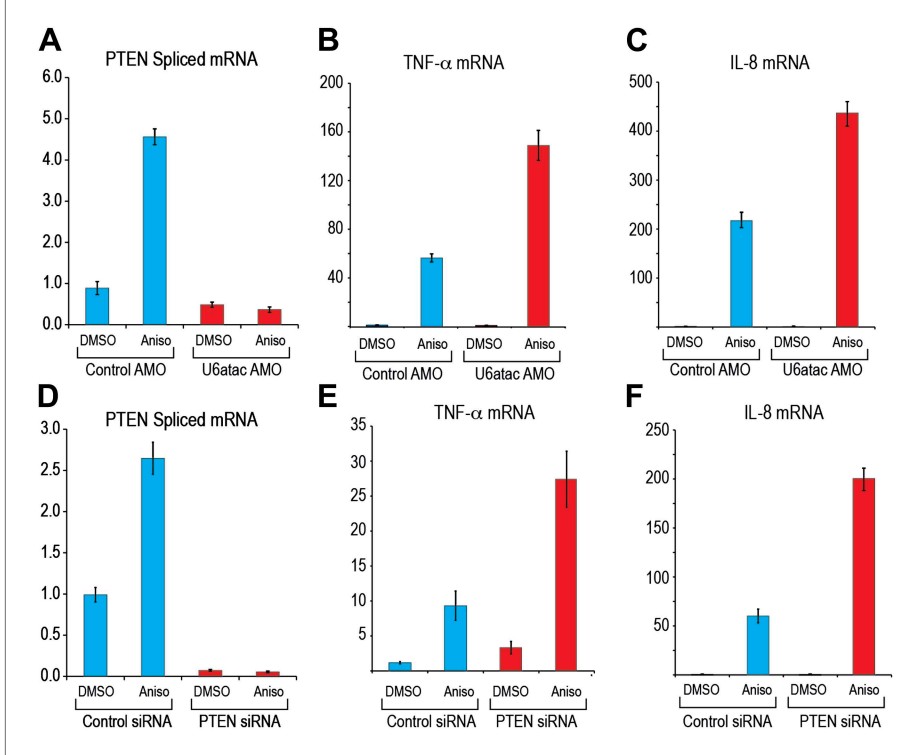

**Figure 6**. U6atac level regulates PTEN minor intron splicing and buffers cytokine production in response to p38MAPK activation. (**A**) Real time qPCR for PTEN in HeLa cells transfected with control or U6atac AMO followed by treatment with DMSO or 1 μg/ml anisomycin for 4 hr. (**B**) and (**C**) Real time qPCR for TNF-α and IL-8 after 4 hr of anisomycin treatment in HeLa cells. (**D**) Real time qPCR for PTEN in HeLa cells transfected with control or PTEN siRNA followed by treatment with DMSO or anisomycin for 4 hr. (**E**) and (**F**) Real time qPCR for TNF-α and IL-8 after 4 hr of anisomycin treatment in HeLa cells transfected with control or PTEN siRNA for 48 hr. Error bars represent the standard deviation of three replicates.

as a specialized post-transcriptional mechanism to regulate expression of their host genes. Minor intron-containing genes that typically contain a single minor intron amidst many major ones function in diverse cellular processes that are critical for cell growth and organism development (*Otake et al., 2002*; *Alioto, 2007*; *Abdel-Salam et al., 2011*; *Edery et al., 2011*; *He et al., 2011*). Fine-tuning the level of the catalytic snRNP, U6atac, allows for a circuit design based on the capacity to completely shut off or rapidly up-regulate the production of the full-length mRNAs from a pool of pre-mRNAs in which all the other major introns have been spliced, without having to affect their transcription (*Figure 7*). Indeed, as our RNA-seq show, a single minor intron is sufficient to regulate the expression of an entire pre-mRNA. The conservation of this design principle over >500 million years (*Burge et al., 1998*; *Shukla and Padgett, 1999*; *Russell et al., 2006*) demonstrates the effectiveness of this gene regulation mechanism. Thus, minor introns function as control switches that are embedded in hundreds of genes and regulated by U6atac abundance, providing a rationale for evolutionary conservation of the minor spliceosome.

## Materials and methods

### RNA preparation and labeling

Total RNAs were prepared from HeLa cells using Trizol (Invitrogen, Grand Island, NY). RNA standards (U11 and U12) were transcribed using T7 MegaScript Kit (Ambion, Grand Island, NY) and gel purified. A modified protocol of the ULYSIS nucleic acid labeling kits (Invitrogen) was used to label RNAs with Alexa546 or Alexa647. Total RNA (1 μg) was combined with 3 pmol each of standard E. coli tRNA$^{Lys}$, *Escherichia coli* tRNA$^{Val}$, and *Saccharomyces cerevisiae* tRNA$^{Phe}$ and resuspended in ULYSIS labeling

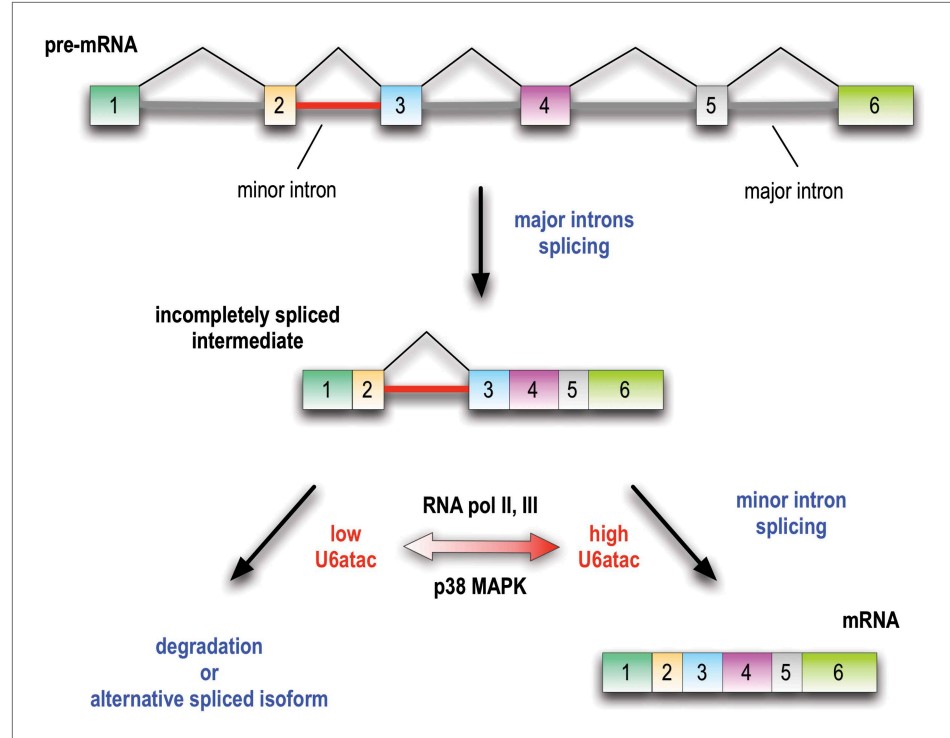

**Figure 7**. Minor introns are embedded molecular switches regulated by U6atac abundance. Splicing of minor introns, typically one amidst several major introns, is dependent on the limiting level of U6atac snRNP in cells. Low abundance and fast turnover of U6atac results in incompletely spliced pre-mRNAs that are either degraded or switch into spliced isoforms that do not require the minor spliceosome. While transcription attenuation rapidly lowers U6atac level and limits minor intron-containing gene's expression, activated p38MAPK rapidly stabilizes U6atac, increasing its level and enhancing minor splicing and production of full-length mRNAs.

buffer (5 mM Tris, 1 mM EDTA, pH 8.0). The RNAs were denatured at 95°C for 5 min, added to either Alexa546 or Alexa647 ULYSIS dye and incubated at 90°C for 10 min. G25 spin columns (GE Healthcare, Pittsburgh, PA) were used to remove excess dye. The labeled RNAs were then precipitated with ethanol for use in microarray hybridization.

## Microarray design, hybridization, and analysis

Two to four 60-mer DNA probes complementary to different regions of the ncRNAs were designed with a GC content between 35% and 55% and limited secondary structure. The probes were spotted on glass slides in quadruplicate at 200 μM on Matrix II slides (Full Moon Biosystems, Sunnyvale, CA) and cross-linked by UV irradiation. Before use slides were pre-hybridized in 2X SSC, 0.2% SDS, 0.1% BSA for 20–30 min, rinsed in water twice and then dried.

Labeled total RNA samples were resuspended in Oligo Hyb Buffer (Full Moon Biosystems, Sunnyvale, CA) containing 10 μg polyA RNA and 20 μg salmon sperm DNA. Samples were applied to the hybridization chamber of the HS4800 Pro hybridzation station (Tecan, Morrisville, NC). The following program for hybridization was used: Wash 1 (54°C, wash 30 s, soak 30 s, 3 runs), two-step Sample Injecion (54°C), Hybridization (54°C, 16 hr), Wash 1 (54°C, wash 30 s, soak 30 s), Wash 1, 2 and 3 (23°C, wash 30 s, soak 30 s), Slide drying (30°C, 5 min). Wash buffers are Wash 1 (2X SSC, 0.2% SDS), Wash 2 (2X SSC), and Wash 3 (0.2X SSC).

Microarray slides were imaged using GenePix 4000b scanner (Axon Instruments, Molecular Devices, Sunnyvale, CA). The fluorescence intensities were quantified and background was subtracted using the GenePix Pro 6.0 software. Values for the 4 replicate spots on each slide were averaged.

Several hybridization temperatures were tested using total RNA and in vitro transcribed snRNAs showing reliable detection of 5 ng of spiked-in U11 and U12 snRNAs without significant

cross-hybridization at temperatures ranging from 48°C to 54°C (*Figure 1—figure supplement 1*). Fluorescence from non-specific probes (red dots) was <10% of the signal of the specific probes, demonstrating the specificity of this array. Similar results were obtained for the measurements of other in vitro transcribed ncRNAs, including U1, U2, U4, U5, U6, U7, U85, U90, U93 and hTR (human telomerase RNA).

The reproducibility between biological replicates was determined on total RNA isolated from three separate HeLa cell cultures, which were labeled with both Alexa546 and Alexa647 and hybridized on the array. For those probes with detectable signals at least twofold above background, the intensity for each probe normalized to the total signal on the chip did not show significant variation among the three biological replicates (*Figure 1—figure supplement 2*, error bars), indicating that the array can reliably discern ncRNA abundance between replicate samples. The dynamic range of the array was determined by testing the linearity of the signals from the different probes with varying amounts of RNA. *Figure 1—figure supplement 1B* shows a range of 50–2500 ng of total RNA labeled with Alexa647 that was hybridized in the presence of 200 ng of Alexa546-labeled total RNA. Most probes exhibited linearity over the range in which experiments are typically performed (100–500 ng total RNA).

## Quantitative real-time PCR

snRNA-specific primers and 5S rRNA-specific primers (endogenous control) were used to generate cDNAs using Transcriptor First-strand cDNA synthesis kit (Roche Applied Sciences, Indianapolis, IN) from an input of 100 ng of total RNA according to manufacturer's instructions. Two and a half percent of the cDNA generated was used for each qPCR reaction (Applied Biosystems 7500 Fast Real-time PCR system) using SYBR Green dye chemistry. The same reverse primers were used for both RT and qPCR. Each reaction was performed in triplicate. For real-time qPCRs and RT-PCRs used to measure splicing of minor and major introns (*Figures 2–6*), 1 µg total RNA was converted to cDNA using the VILO kit (Invitrogen) according to manufacturers recommendations. cDNA was then diluted to 10 ng/µl and 20–50 ng cDNA was then used as input for qPCR (with SYBR Green as described above) or regular PCR using platinum Taq (Invitrogen). For qPCR, all the primer sets for various genes span the splice junction and thus measure only spliced mRNAs. For RT-PCR, the primers are located in the surrounding exons and thus amplify both spliced and unspliced mRNAs.

## Cell treatments, library preparations and RNA-seq

HeLa cells were grown in DMEM supplemented with 10% fetal bovine serum (FBS), 1% antibiotics and maintained at 37°C in a 5% $CO_2$ humidified atmosphere. Actinomycin D (5 µg/ml), cycloheximide (10 µg/ml), DRB (100 µM), nocodazole (200 nM), or anisomycin (1 µg/ml) were added to the media for the indicated time, followed by RNA extraction as indicated above.

Luciferase splicing reporters were generated as previously described (*Younis et al., 2010*). Briefly, the minor intron of the CHD4 gene was inserted at nucleotide position 571 of firefly luciferase gene. The luciferase protein is destabilized using both a PEST protein degradation sequence as well as a CL1 sequence, and mRNA is destabilized by adding five tandem AUUUA repeats into the 3′UTR. Both intron-containing and intronless luciferase were transcribed from a CMV promoter.

Control (5′-CCT CTT ACC TCA GTT ACA ATT TAT A-3′) and U6atac (5′-AAC CTT CTC TCC TTT CAT ACA ACA C-3′) antisense morpholinos (AMOs) were transfected into cells using the Neon system (Invitrogen). 6 hr post transfection, cells were labeled with 200 µM 4-thiouridine (4-SU) for an additional 2 hr. Total RNA was extracted using Trizol (Invitrogen) followed by extraction of polyA-containing RNA on oligo-dT columns using Oligotex kit (Qiagen, Germantown, MD). For isolation of 4-SU-labeled nascent transcripts, EZ-Link biotin-HPDP (Thermo Scientific, Waltham, MA) was reacted with the 4-SU and purified on streptavidin Dynabeads (Invitrogen) as previously described (*Dolken et al., 2008*).

Nascent transcripts prepared as described above were used to prepare cDNA libraries using Encore NGS Library System I (Nugen, San Carlos, CA) according to the manufacturer's recommendations. Briefly, 100 ng RNA was converted to cDNA and amplified using the Ribo-SPIA technology. The cDNA was then fragmented, end repaired, ligated to Illumina adaptors and bead purified according to instructions in the Ovation RNA-seq System (Nugen). Sequencing was performed at the University of Pennsylvania Core Facility on Illumina Hi-Seq2000 platform to generate single end 100 base reads. Raw and processed data is available on GEO under the accession number GSE48263. This link can be used to retrieve the data: http://www.ncbi.nlm.nih.gov/geo/query/acc.cgi?acc=GSE48263.

## Data analysis and statistics

RNA-seq reads were aligned to the reference genome (UCSC, hg19) using Tophat by default parameters (*Trapnell et al., 2009*). Uniquely mapped reads and the best of multiple hits were kept for downstream analysis. BEDTools software (*Quinlan and Hall, 2010*) was used to generate reads per exon, from which FPKM (read fragments per kilobase of exon model per million mapped reads) values were calculated (*Mortazavi et al., 2008*). The analysis of differential gene expression was performed using Cufflinks (*Trapnell et al., 2010*). Significant genes were detected based on FDR level at 0.05 using Benjamini-Hochberg correction for multiple-testing (*Trapnell et al., 2012*).

To identify differentially expressed splicing isoforms across samples and quantify the expression level of those alternative spliced genes, MISO (version 0.4.1 with default parameters) was applied to the aligned RNA-seq data (*Katz et al., 2010*). Briefly, MISO estimates expression at alternative splicing event level by computing PSI (Percent Spliced Isoform) and measures the differential expression by Bayes factors. The MISO results were filtered for alternative splicing events using the following criteria: (a) at least 15 inclusion read, (b) 15 exclusion read, such that (c) the sum of inclusion and exclusion reads is at least 30, and (d) the $\Delta\Psi$ is at least 0.30, and (e) the Bayes factor is at least 10, and that (a)–(e) are true in one of the samples.

## RNase H protection assay

Total cell extract was prepared from AMO-transfected cells using 10 mM Tris-HCl pH 7.5, 2.5 mM MgCl$_2$, 100 mM NaCl and 0.1% NP-40. RNase H along with 5 µM antisense DNA oligonucleotide for U6atac (5′-TCA TAC AAC AC-3′) was added for 25 min at 30°C, and RNA was purified and analyzed by northern blotting with a U6atac snRNA probe (5′-CCG TAT GCG TGT TGT CAG GCC CGA GGG CCT-3′).

## Acknowledgements

We are grateful to members of our laboratory, especially Dr Anna Maria Pinto, for comments on this manuscript. We thank the microarray facility at the University of Pennsylvania School of Medicine for printing the ncRNA probes and the Functional Genomics Core for Illumina sequencing. GD is an Investigator of the Howard Hughes Medical Institute.

## Additional information

### Funding

| Funder | Author |
| --- | --- |
| Howard Hughes Medical Institute | Gideon Dreyfuss |
| Association Française Contre les Myopathies | Gideon Dreyfuss |

The funders had no role in study design, data collection and interpretation, or the decision to submit the work for publication.

### Author contributions

IY, Conception and design, Acquisition of data, Analysis and interpretation of data, Drafting or revising the article; KD, Conception and design, Acquisition of data; WW, ZW, Analysis of data, Drafting the article; SWF, KYH, Revising the article, Acquisition of data; MGB, Interpretation of data, Drafting or revising the article; LW, GD, Conception and design, Analysis and interpretation of data, Drafting or revising the article

## Additional files

### Supplementary files

• Supplementary file 1. (**A**) Noncoding RNAs represented on the microarray. (**B**) Minor intron-containing genes with lowest and highest splicing indices and their function.

## Major dataset

The following dataset was generated:

| Author(s) | Year | Dataset title | Dataset ID and/or URL | Database, license, and accessibility information |
|---|---|---|---|---|
| Dreyfuss G, Younis I | 2013 | Genome wide mapping of effects of U6atac knockdown on pre-mRNA splicing | GSE48263; http://www.ncbi.nlm.nih.gov/geo/query/acc.cgi?acc=GSE48263 | Publicly available at GEO (http://www.ncbi.nlm.nih.gov/geo/). |

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
