## [Decision Letter]

Thank you for sending your work entitled “Minor introns are embedded molecular switches regulated by highly unstable U6atac snRNA” for consideration at *eLife*. Your article has been favorably evaluated by a Senior editor and 3 reviewers, one of whom is a member of our Board of Reviewing Editors.

The Reviewing editor, Timothy Nilsen, and the other reviewers discussed their comments before we reached this decision, and the Reviewing editor has assembled the following comments to help you prepare a revised submission.

Your manuscript has been reviewed by three experts in the area of regulated alternative splicing and splicing by the “minor” spliceosome. In general, the reviewers were quite positive and there is a consensus that the work is potentially acceptable for publication in *eLife* because it provides an interesting rationale for retention of the minor spliceosome as a complex regulatory machine.

Nevertheless there are a few issues that need to be addressed before the manuscript can be reconsidered.

1) Previous reports have indicated that U4atac is the least abundant of the minor spliceosomal snRNAs and that there was a considerable excess of free U6atac. Thus, the reviewers would like to see quantitation of the levels of U4atac/U6atac, and U4atac under the conditions you describe. Similarly, it would be important to quantify snRNA levels (and di-snRNP) in the anisomycin experiments.

2) There is a concern that the effects measured on TNFα and IL8 may not be a direct consequence of activation (in anisomycin) or repression (U6atac sequestration) of PTEN. This concern derives from the fact that knockdown of PTEN in the absence of anisomycin does not affect cytokine expression. It is suggested that PTEN expression be assayed directly by Western blot as well as measuring phosphorylation status of direct PTEN targets under the conditions described.

---

## [Author Response]

*1) Previous reports have indicated that U4atac is the least abundant of the minor spliceosomal snRNAs and that there was a considerable excess of free U6atac. Thus, the reviewers would like to see quantitation of the levels of U4atac/U6atac, and U4atac under the conditions you describe. Similarly, it would be important to quantify snRNA levels (and di-snRNP) in the anisomycin experiments*.

In light of this comment, we measured the amount of U4atac, U6atac, and U4atac/U6atac di-snRNP in the same HeLa cells used in experiments throughout this manuscript by real-time absolute quantification. Indeed, we found U4atac to be even less abundant than U6atac (∼20-fold). Interestingly U4atac levels measured from total RNA decreased by ∼20% after 4 hours of Actinomycin D treatment and did not change after 4 hours of anisomycin treatment. On the other hand, the level of U4atac/U6atac that is associated with the di-snRNP specific protein p110/SART3 was significantly decreased after Actinomycin D treatment, which mirrors the decrease of U6atac in total RNA. Importantly, anisomycin treatment caused the p110/SART3-associated U4atac/U6atac di-snRNP to increase. These findings indicate that U6atac is the rate-limiting factor for U4atac/U6atac di-snRNP formation and minor intron splicing. We have added these data as Figure 1—figure supplement 3 and describe them in the text.

*2) There is a concern that the effects measured on TNFα and IL8 may not be a direct consequence of activation (in anisomycin) or repression (U6atac sequestration) of PTEN. This concern derives from the fact that knockdown of PTEN in the absence of anisomycin does not affect cytokine expression. It is suggested that PTEN expression be assayed directly by Western blot as well as measuring phosphorylation status of direct PTEN targets under the conditions described*.

These are constructive comments and we have given them much thought, followed by several experiments. Regarding the lack of an effect of PTEN knockdown on cytokine expression in the absence of anisomycin: without a cytokine inducer (e.g., the p38MAPK activator anisomycin), the baseline expression level of these cytokines in HeLa cells is too low to determine if PTEN knockdown has an effect on them.

To further explore the relationship between TNF-a and IL8 down-regulation and U6atac-mediated PTEN up-regulation, we have measured PTEN and cytokines protein levels in two different cell lines (HeLa and THP-1) treated with anisomycin, which activates p38MAPK and increases/stabilizes U6atac. Cytokine proteins’ increase reflected those of their mRNAs (>10 fold) in both cell lines, and was increased further by U6atac AMO. On the other hand, the amount of PTEN protein increase was ∼20%, which we are not confident is sufficient to explain this protein’s role in the observed cytokines’ response. We note, however, that similarly small changes in PTEN protein levels have been shown to have significant biological effects, including on cancer development in mice (Alimonti et al, Nature Genetics, 2010). Thus, while U6atac level can modulate PTEN mRNA and similarly the cytokines’ mRNAs and proteins over a several fold range, the direct U6atac-dependent regulatory factor(s) remains to be determined. However, given PTEN’s key role in human pathologies, the potential of U6atac modulation to regulate PTEN should be considered. We have added these points to the text.

It is also possible that the cytokine repression is mediated by a mechanism that depends on PTEN mRNA rather than the protein product. We also cannot rule out that anisomycin, known to also inhibit protein synthesis typically at higher concentration, dampens the effect on protein level.